# The Effect of Lithium Ion Leaching from Calcined Li–Al Hydrotalcite on the Rapid Removal of Ni^2+^/Cu^2+^ from Contaminated Aqueous Solutions

**DOI:** 10.3390/nano13091477

**Published:** 2023-04-26

**Authors:** Yu-Jia Chen, Jun-Yen Uan

**Affiliations:** 1Department of Materials Science and Engineering, National Chung Hsing University, Taichung 402, Taiwan; t820207@gmail.com; 2Innovation and Development Center of Sustainable Agriculture (IDCSA), National Chung Hsing University, Taichung 402, Taiwan; 3Industrial and Intelligent Technology Degree Program, Academy of Circular Economy, National Chung Hsing University, Taichung 402, Taiwan

**Keywords:** layered double hydroxides, lithium leaching, structured adsorbents, metal recovery, wastewater treatment, nanoparticles

## Abstract

A layered double hydroxide (LDH) calcined-framework adsorbent was investigated for the rapid removal of heavy metal cations from plating wastewater. Li–Al–CO_3_ LDH was synthesized on an aluminum lathe waste frame surface to prepare the sorbent. The calcination treatment modified the LDH surface properties, such as the hydrophilicity and the surface pH. The change in surface functional groups and the leaching of lithium ions affected the surface properties and the adsorption capacity of the heavy metal cations. A zeta potential analysis confirmed that the 400 °C calcination changed the LDH surface from positively charged (+10 mV) to negatively charged (−17 mV). This negatively charged surface contributed to the sorbent instantly bonding with heavy metal cations in large quantities, as occurs during contact with wastewater. The adsorption isotherms could be fitted using the Freundlich model. The pseudo-second-order model and the rate-controlled liquid-film diffusion model successfully simulated the adsorption kinetics, suggesting that the critical adsorption step was a heterogeneous surface reaction. This study also confirmed that the recovered nickel and/or copper species could be converted into supported metal nanoparticles with a high-temperature hydrogen reduction treatment, which could be reused as catalysts.

## 1. Introduction

Nickel and copper nanocatalysts have a wide range of catalytic applications because they are relatively low-cost and catalytically efficient [1,2,3]. Nickel plating and copper plating are frequently used industrially for the surface treatment of workpieces [4,5]. Significant quantities of nickel/copper cations are frequently present in plating wastewater. The discharged wastewater containing nickel and copper ions causes harm to the human body and the environment. The most common harm to the human body caused by nickel is contact dermatitis, which causes skin allergies and inflammation; it can also cause eczema [6,7]. Copper enters the human body mostly through drinking water [8]. Copper causes major hazards for the human body, including high blood pressure, a high respiratory rate, liver and kidney failure, and convulsions [6,7,8]. Several methods have been developed for removing heavy metal ions from wastewater, such as chemical precipitation [9], adsorption [10], ion exchange [11], membrane filtration [12], and electrodialysis [13].

Adsorption is an efficient and frequently used method for removing toxic heavy metal ions from wastewater due to its low cost and simple process compared with other technologies [14,15,16]. The general requirements for newly used adsorbents include low cost, high adsorption capacity, ease of handling, and simple design [7]. The disposal of used adsorbents containing heavy metals remains a problem, the resolution of which may impose additional costs and cause secondary pollution of the environment [17,18]. An economical and environmentally friendly approach is the recovery of nickel/copper cations from plating wastewater for use in practical applications, such as their usage as catalysts. Layered double hydroxides (LDHs) and their derivatives are materials with the potential for use in the recovery and loading of heavy metals due to their high specific surface area, the high density of their surface functional groups, and their thermal stability [19,20,21]. LDHs, which are also called hydrotalcites, are groups of clays with a layered structure comprising positively charged layers with intercalated water molecules and anions for charge balance [22,23,24]. The positively charged layers comprise monovalent, divalent, or trivalent metal cations in hydroxyl slabs. Li–Al LDHs are unique LDHs that contain the monovalent metal cation Li^+^, occupying some of the vacant octahedral sites of gibbsite (Al(OH)_3_) sheets to form positively charged layers [25,26,27]. As shown in the authors’ previous work [28], the Li–Al–CO_3_ LDH has a typical chemical formula of Li_0.33_Al_0.67_(CO_3_)_0.17_(OH)_2_·*m*H_2_O. The authors’ previous studies [29,30] prepared calcined Li–Al LDH loaded with Ni and/or Cu nanoparticles and used it in the catalytic reaction of ethanol steam, reforming to produce hydrogen. However, the process of quickly absorbing Ni^2+^/Cu^2+^ from wastewater by the calcined Li–Al LDH is still unclear. It is this process that is investigated in this study.

The calcination of LDHs is one of the methods for modifying their properties and improving their performance in adsorption [31,32,33]. The calcination of LDHs removes interlayer water molecules and anions, as well as the hydroxyl groups, resulting in the loss of the layered structure [34]. The calcination of LDHs at high temperatures typically takes several hours (2–6 h at 400–550 °C) [35,36,37,38,39,40,41]. Furthermore, the heating of LDH powder uniformly is challenging, due to its low thermal conductivity. To improve the efficiency of the removal of heavy metal ions from wastewater, a large amount of calcined LDH powder is put into the water. The adsorbents may aggregate during adsorption, reducing the efficiency of adsorption and retrieval [42]. Powdered sorbents must be separated from the treated wastewater by centrifugation or precipitation after the adsorption of heavy metal ions. These processes are usually lengthy and consume considerable energy. Therefore, to address these problems, this research focused on heating the LDHs uniformly by coating them on a metallic structured frame surface. Aluminum lathe waste, an industrial residue generated in the processing of aluminum alloys on lathes, was used as the structured frame for coating the Li–Al LDH films.

Possible mechanisms for recovering heavy metal ions by calcined LDHs include surface complexation, precipitation, isomorphous substitution, and coprecipitation [40,41,43]. Sun et al. [40] posited that the metal cations (Ni^2+^, Cu^2+^, and Zn^2+^) first precipitated on the surface of the adsorbent (calcined Mg–Al LDH) to form hydroxides, which then participated in the process by which the adsorbent reconstructs the LDH structure through isomorphous substitution. Yang et al. [41] proposed possible mechanisms for removing Ni^2+^/Cu^2+^ by calcined Mg–Al LDH-modified palygorskite adsorbents through surface complexation, coprecipitation, and isomorphous substitution. It is well known that calcined LDH may regenerate into an LDH structure when exposed to water and anions [34]. The regeneration process usually takes several hours [44,45] as anions and water molecules enter the interlayer, reconstructing the LDH-layered structure. Wong and Buchheit [45] found that Li–Al LDH calcined at 220 °C had to be immersed for five days to reconstruct the LDH structure completely. Rocha et al. [44] found that Mg–Al LDH calcined at 550 °C or below took 24 h of rehydration to complete reconstruction. The isomorphous substitution of metal cations into LDHs also takes several hours [46,47]. Kim et al. studied the isomorphous substitution of divalent metal ions into LDH, finding that it took six hours to substitute ~27% of the Mg^2+^ into Mg–Al LDH using Co^2+^ [46]. Richardson and Braterman [47] found that the substitution of Ni^2+^/Co^2+^/Zn^2+^/Cu^2+^/Mn^2+^ into Mg–Al LDH, which had M(Ⅱ):Mg atomic ratios ranging from 0.61 to 1.25, took five days. However, based on the adsorption experiments of Sun et al. [40] and Yang et al. [41], the removal of Ni^2+^/Cu^2+^ took a relatively short time (30 min) before reaching their removal equilibrium capacity, indicating that the reconstruction of the calcined LDH and the isomorphous substitution of metal cations may not be the controlled mechanisms of the quick metal cation recovery.

The present research investigated the effects of LDH calcination temperature and reaction conditions on recovery efficiency to elucidate the mechanism of the quick uptake of heavy metal cations from wastewater. Moreover, instead of conventionally leaching Li^+^ in strong acids with the addition of a reducing agent from spent lithium-ion batteries [48], this research investigated the novel approach of Li^+^ leaching in weak acid (pH 4–6) without using a reducing agent for cation recovery.

## 2. Materials and Methods

### 2.1. Synthesis of Li–Al–CO_3_ LDH on Aluminum Lathe Waste

An industrial 6061 aluminum lathe waste was obtained from Fullchamp Technologies Co., Ltd., Nantou County, Taiwan. Figure 1a,b show photographs of the aluminum lathe waste. The aluminum lathe waste was ultrasonically cleaned in acetone to remove oil and debris from its surface. A 1000 mL aqueous solution that contained Al^3+^ and Li^+^, with a Li/Al molar ratio of two, was prepared by adding strips of aluminum foil to a 0.06 M LiOH aqueous solution. This solution was magnetically stirred at 50 °C for 30 min and purged with highly pure Ar gas to reduce the dissolution of contaminating CO_2_. It was then filtered through filter paper to obtain a clear solution. A 10 g mass of aluminum lathe waste was dipped in the solution at room temperature for 2 h to form the Li–Al–CO_3_ LDH film coated on the aluminum lathe waste surface. The aluminum lathe waste, coated with Li–Al–CO_3_ LDH, was naturally air-dried. The LDH was then calcined in the air at various temperatures (150–500 °C) for 1 h to obtain the calcined LDH. Herein, the sample of 150 °C-calcined LDH was denoted as T150; and the samples of 200 °C, 300 °C, 400 °C, and 500 °C-calcined LDH were denoted as T200, T300, T400, and T500, respectively.

### 2.2. Ni^2+^ and Cu^2+^ Uptake from Aqueous Solutions

NiSO_4_·6H_2_O (99%, Choneye Pure Chemicals, Miaoli County, Taiwan) and CuSO_4_·5H_2_O (98%, Choneye Pure Chemicals, Miaoli County, Taiwan) were used to prepare the aqueous solutions that contained Ni^2+^ and Cu^2+^, respectively. Adsorption experiments were performed by adding the sorbent (the sorbent coated on lathe waste, as shown in Figure 1a) to the aqueous solutions that contained Ni^2+^/Cu^2+^ at 25 °C with stirring at 600 rpm. The concentrations of the cations were measured using an inductively coupled plasma optical emission spectrometer (ICP-OES; Agilent 5110 ICP-OES, Santa Clara, CA, USA). The measurements were repeated three times. The removal efficiency (*R*) was calculated as follows:R = ((C_0_ − C_e_)/C_0_) × 100%(1)
where C_0_ (mg/L) and C_e_ (mg/L) are the initial and equilibrium concentrations of the metal cations, respectively. The adsorption capacity (Q(t)) was calculated as follows:Q(t) (mg/g) = ((C_0_ − C_e_) × V)/m(2)
where V (L) is the solution volume and m (g/L) is the adsorbent dosage.

Moreover, aqueous solutions containing Ni^2+^ were used to study the effect of LDH calcination temperature on the removal efficiency of heavy metal cations. Samples of 1 and/or 2 g of LDH/aluminum lathe waste were calcined at various temperatures for 1 h for use in each adsorption experiment. Adsorption was performed in 100 mL aqueous solutions for 1 h with an initial Ni^2+^ concentration of 75 mg/L. To investigate the adsorption kinetics, calcined LDH was added to 400 mL of aqueous solution with an initial Ni^2+^/Cu^2+^ concentration of 75 mg/L. Sample solutions were taken at intervals to measure the Ni^2+^/Cu^2+^ concentration. Adsorption isotherm studies were performed in 100 mL aqueous solutions with initial Ni^2+^/Cu^2+^ concentrations of 75, 120, 240, 360, 480, and 600 mg/L. The effect of solution pH on the formation of the Ni/Cu products was studied. T400 was used to investigate the effect of solution pH on Ni/Cu products. The products are denoted as Ni-pH4.0, Ni-pH6.3, Ni-pH7.5 and Cu-pH3.9, Cu-pH5.5, and Cu-pH7.5, indicating the metal ions in the aqueous solution and the initial solution pH. A 500 mL volume of an aqueous solution that contained 75 mg/L of Ni^2+^/Cu^2+^ was added dropwise to aqueous NaHCO_3_ or H_2_SO_4_ to adjust its initial solution pH. Recovery was performed with very low sorbent/aqueous solution ratios to minimize the variation of the solution pH. Li–Al–CO_3_ LDH was synthesized on aluminum sheets measuring 20 × 20 × 2 mm^3^ by the method that was described in Section 2.1 for use in experiments. The calcined LDH/aluminum sheets were used to recover Ni^2+^/Cu^2+^ for one hour. The used calcined LDH, after recovering Ni^2+^/Cu^2+^, was recycled to be prepared as nano-catalysts. For this purpose, the used calcined LDH was reduced with highly pure H_2_ at a 100 mL/min flow rate in a tubular furnace at 500 °C.

### 2.3. Characterization

A field emission scanning electron microscope (FE-SEM; Zeiss Ultra Plus, Oberkochen, Germany) was used to observe the surface morphologies of the samples. Prior to the SEM observation, the samples were coated with platinum. The surface elemental composition was determined using an energy-dispersive X-ray spectroscope (EDS; Oxford X-Max 50 mm^2^, Oxford, UK) attached to the SEM with an accelerating voltage of 7 kV. A field emission transmission electron microscope (FE-TEM; FEI Tecnai G2 F20, Hillsboro, OR, USA) was used to observe the cross-sectional microstructures of the samples. The TEM also obtained the high-angle annular dark field (HAADF), high-resolution transmission electron microscopy (HRTEM) images. Specimens for TEM observation were prepared using a focused ion beam (FIB; Hitachi NX2000, Tokyo, Japan). A carbon film was deposited on each sample before FIB bombardment to protect the sample’s surface from damage. The crystallographic structures of the samples were determined by glancing angle X-ray diffraction (GAXRD; Bruker D8 Advance ECO, Darmstadt, Germany) with a glancing angle of 0.5° using Cu Kα_1_ (1.5406 Å) radiation. Fourier-transform infrared (FTIR; PerkinElmer Spectrum 65 FT-IR spectrometer, Norwalk, CT, USA) spectra were obtained in the wavenumber range 450–4000 cm^−1^ with a resolution of 4 cm^−1^ and eight scans, using the attenuated total reflection (ATR) method. X-ray photoelectron spectroscopy (XPS; ULVAC-PHI PHI 5000 VersaProbe, Kanagawa, Japan) was used to detect the chemical states of surface elements. Prior to the XPS examination, the surface of each sample was bombarded with Ar ions for 6 s to remove surface-adsorbed gas. To investigate the surface properties of the as-prepared and calcined LDH, the water contact angle was measured using a contact angle analyzer (First Ten Angstroms FTA-2000, Portsmouth, VA, USA). Surface pH was measured using a flat surface pH electrode (Hamilton FlaTrode, Bonaduz, Switzerland). For each pH measurement, 0.1 mL of deionized water was dropped onto the sample surface, to which a pH electrode was attached. The zeta potential of LDH was compared with that of the gibbsite to determine the role of Li^+^. The Li–Al LDH powder was scraped from the aluminum substrates. The extra pure reagent-grade gibbsite powder was purchased from Choneye Pure Chemicals, Miaoli County, Taiwan. The Li–Al LDH, gibbsite, and their calcination products were added to pH 5.0 weakly acidic water (titrated with sulfuric acid) to measure the zeta potential. The calcination of Li–Al LDH and gibbsite was carried out at 400 °C for 1 h. The calcined gibbsite herein was denoted as C-gibbsite. Additional T400 powder was added to aqueous solutions of NiSO_4_/CuSO_4_ (concentrations of Ni^2+^/Cu^2+^ = 75 mg/L), denoted as T400-Ni and T400-Cu, respectively. The ratio of powder to the aqueous solution was 0.01 g/20 g. The zeta potential was measured using a Litesizer 500 (Anton Paar, Graz, Austria) with the electrophoretic light scattering (ELS) method, which measures the speed of particles as an electric field is applied. The zeta potential was measured at 1, 10, and 60 min after adding the powder to the aqueous solution.

## 3. Results and Discussion

### 3.1. Observation of the Prepared Adsorbent Material

Figure 1 presents macroscopic and microscopic views of the LDH adsorbent material. Figure 1a shows a photograph of a pile of industrial aluminum lathe waste, from which the substrates for forming Li–Al–CO_3_ LDH were obtained. Figure 1b displays an enlarged view of a strip of aluminum lathe waste that had been coated with Li–Al–CO_3_ LDH. Figure 1c shows the SEM surface morphologies of the Li–Al–CO_3_ LDH that was coated on the aluminum lathe waste substrate. The Li–Al–CO_3_ LDH had vertical platelet-like structures and densely covered the surface of the aluminum substrate. The inset in Figure 1c displays a high-magnification image of the LDH platelets. The platelets were between 1 and 2 μm long and about 20–40 nm wide. Figure 1d displays a cross-sectional TEM image of the LDH. The LDH platelets were about 3 μm high and had a bottom layer of precursors (~0.5 μm thickness attached to the aluminum substrate).

### 3.2. The Surface Characterization of Calcined LDH

Figure 2 displays the surface characteristics of the LDH and calcined LDH samples. Figure 2a shows the XRD patterns of the samples. As shown in Figure 2a, the pristine LDH and T150 exhibit a prominent Li–Al–CO_3_ LDH (002) characteristic peak at 2θ = 11.65°; the characteristic peaks at planes of (101), (004), and (112) are also shown. The characteristic peaks of Li–Al–CO_3_ LDH following calcination at 200 °C were less intense than that of the pristine LDH and T150. These characteristic peaks disappeared with calcination at a temperature above 300 °C, and there was no apparent other peak formation, indicating that the structure had transformed into an amorphous form. The Li–Al LDH was transformed into an amorphous mixed-metal oxide phase by calcination at 300–500 °C [45,49,50]. Figure 2b displays the FTIR spectra of the LDH and calcined LDH samples. In the spectrum of the LDH (bottom curve) in Figure 2b, the broad transmission band at ~3470 nm^−1^ arises from the H-bonding stretching vibration of the O–H groups (ν_O–H_) in the metal hydroxide layers [51]. The intensity of this band decreases as the calcination temperature increases significantly above 300 °C. The decrease in the intensity of the ν_O–H_ band with increasing temperature indicates that calcination destroyed the metal hydroxide layers of LDH. The results supported that the intensity of M–OH (ν_M–OH_ = ~1000 cm^−1^) and Al–OH groups (ν_Al–OH_ = ~700 cm^−1^) decreased with increasing calcination temperature. The band corresponding to the bending mode of interlayer water molecules (δ_H2O_) at 1630 nm^−1^ [52] disappears upon calcination above 300 °C, indicating the escape of water molecules from the interlayers at high temperatures. Similarly, the bands at ~1550, 1370, and 870 cm^−1^, which characterize the asymmetric stretching modes of CO_3_^2−^ [52], weaken as the calcination temperature increases and disappear upon calcination above 300 °C, revealing the removal of the intercalated CO_3_^2−^ groups. The FTIR examination results showed that the structure of LDH was significantly changed by calcination at temperatures above 300 °C.

Figure 2c displays XPS O 1s spectra of the LDH and calcined LDH samples. A calcined LDH sample calcined at 500 °C for 3 h (herein denoted as Li–Al–O) was used to identify the O 1s state of metal oxide. The O 1s XPS spectra in Figure 2c peak at 530.7, 531.7, and 534.0 eV, which are attributed to the metal–oxygen bonds (M–O), hydroxyl groups (–OH), and adsorbed water (H_2_O), respectively [53,54,55]. The LDH (the bottom curve in Figure 2c) exhibits a major peak of the M–OH bond at a binding energy of 531.7 eV. The Li–Al–O sample (top curve) yields a major peak associated with the M–O bond at a binding energy of 530.7 eV, indicating that calcination at 500 °C for 3 h can convert almost all M–OH bonds to M–O bonds. The H_2_O in the calcined LDH is insignificant in calcination above 300 °C. Thus, the LDH calcined for one hour at various temperatures had various M–OH and M–O compositions. Figure 2d plots the M–O molar percentages of the various calcined LDH samples, as determined from the XPS spectra in Figure 2c. The M–O molar percentage increased with temperature, rising from ~15% for LDH to ~81% for T500. The Li–Al–O had an M–O molar percentage of ~85% (not shown in Figure 2d). The evolution of surface hydroxyl groups (denoted as –OH) to oxygen species (denoted as –O–) during calcination can be expressed as the following dehydration reaction [50,56]:–OH + –OH → –O– + H_2_O(3)

Figure 2e shows the water contact angles of the LDH and calcined LDH samples. The water contact angle was significantly reduced from ~116° for pristine LDH to less than 10° by calcination at various temperatures, especially that of T400 being reduced to 53°. The inset in Figure 2e shows the significant difference in water contact angle between the pristine LDH and calcined LDH T400, confirming that calcination treatment changed the LDH film from hydrophobic to hydrophilic. Figure 2f shows the surface pH values of the LDH and calcined LDH. The pristine LDH had a surface pH of ~9.5. As the calcination temperature increased, the surface pH gradually increased. The surface pH reached ~10.75 at a calcination temperature of 400 °C, with no noticeable increase to 500 °C.

### 3.3. The Effect of LDH Calcination Temperature on Removal Efficiency

Figure 3a compares the Ni^2+^ removal efficiencies of the LDH and calcined LDH samples. As shown in Figure 3a, when 1 g of LDH/aluminum lathe waste was used, neither pristine LDH nor T150 had any capacity to remove Ni^2+^. The calcined LDH exhibited an enhanced ability to recover Ni^2+^ as the calcination temperature increased. As the calcination temperature increased to 300 °C, the removal efficiency was significantly increased to ~35%. At calcination temperatures of 300–500 °C, the removal efficiency of Ni^2+^ increased as the temperature increased, reaching ~42% at 500 °C. Using 2 g of LDH/aluminum lathe waste yielded a much higher removal efficiency of Ni^2+^ than 1 g at each calcination temperature, significantly above 300 °C. The Ni^2+^ removal efficiencies obtained using 2 g of LDH/aluminum lathe waste that had been calcined at 300, 400, and 500 °C exceeded 93%. LDH samples calcined at 300, 400, and 500 °C varied negligibly in recovery efficiency.

Figure 3b plots the leaching concentrations of Al^3+^ and Li^+^ of the LDH and calcined LDH samples during the recovery of Ni^2+^. As shown in Figure 3b, the LDH and calcined LDH samples had leaching concentrations of Al^3+^ of less than 2.5 mg/L, indicating that the leaching of Al^3+^ was insignificant. The Li^+^ leaching concentration generally increased with calcination temperature, from ~2.0 mg/L for LDH to ~7.2 mg/L for T300 and T400; it decreased to ~5.7 mg/L for T500. Lee and Jung [57] confirmed that the amount of Li^+^ released from calcined Li–Al LDH in pure water significantly increased with LDH calcination temperature above 300 °C. Based on first-principles calculations by Zhang et al. [58], the Li–O bonds in Li–Al LDH become weaker as the calcination temperature increases above ~200 °C. The release of Li^+^ in water is evidence of the weakening of the Li–O bonds. The release of Li^+^ by the breakage of Li–O bonds in water can be expressed as follows:–O–Li + H_2_O → –O–H + OH^−^_(in solution)_ + Li^+^_(in solution)_(4)

The reaction of Equation (4) generates OH^−^, increasing the solution pH. As previously shown, Figure 2f and Figure 3b confirmed the reaction of Equation (4) from left to right concerning the surface pH and the Li^+^ leaching to increase the Li^+^ concentration in the aqueous solution. As there are heavy metal cations in the aqueous solution, the oxygen species attract heavy metal cations as follows:–O–Li + Me^2+^_(in solution)_ + H_2_O → –O–Me–OH + H^+^_(in solution)_ + Li^+^_(in solution)_(5)

Heavy metal cations bonded to surface oxygen species can electrostatically attract OH^−^ in aqueous solutions, possibly forming hydroxides. Figure 4 shows SEM surface morphologies following the removal of Ni^2+^ for 1 min using the LDH and calcined LDH samples. The LDH, T150, and T200 (Figure 4a–c) had bare reaction products on the platelet surfaces. In contrast, T300 and T400 (Figure 4d,e) had dense cross-linked meshwork structures that formed on the platelet surfaces. For T500 (Figure 4f), the reaction products were irregular aggregations on the platelet surfaces. LDH that had been calcined at 400 °C was used in the subsequent studies. The calcination of LDH reduced its mass. The calcined LDH accounted for approximately 0.04 g per 2 g of the T400 calcined LDH/Al framework.

### 3.4. The Kinetics of Ni^2+^/Cu^2+^ Recovery

The recovery of Ni^2+^/Cu^2+^ using T400 was performed to study its adsorption kinetics. The experimental data were fitted using four typical adsorption kinetic models: pseudo-first-order (Equation (6)), pseudo-second-order (Equation (7)), intraparticle diffusion (Equation (8)), and liquid film diffusion (Equation (9)), as follows [59,60]:
Pseudo-first-order model:    Q(t) = Q_e_(1 − exp(−k_1_t))(6)
Pseudo-second-order model:    Q(t) = k_2_Q_e_^2^t/(1 + k_2_Q_e_t)(7)
Intraparticle diffusion model:    Q(t) = k_id_t^0.5^ + C(8)
Liquid film diffusion model:    ln(1 − (Q(t)/Q_e_)) = −k_fl_t + C(9)
where t (min) is the adsorption time; Q(t) (mg/g) is the adsorption amount at time t; Q_e_ (mg/g) is the equilibrium adsorption capacity; k_1_ (min^−1^), k_2_ (g/mg/min), k_id_ (mg/g/min^0.5^) and k_fd_ (min^−1^) are the rate constants of pseudo-first-order, pseudo-second-order, intraparticle diffusion, and liquid film diffusion models, respectively; C (mg/L) is a constant that is related to the thickness of the boundary layer.

Table 1 lists the kinetic parameters of the removal of Ni^2+^ and Cu^2+^, and the fitting results following the kinetic parameters are plotted in Figure 5. Figure 5a plots the variation of Ni^2+^ concentration with time. The concentration of Ni^2+^ decreased rapidly within the first ten minutes from the initial 75 mg/L to ~35 mg/L. Subsequently, it decreased slowly to ~8 mg/L at 60 min and 0.5 mg/L at 300 min. The inset of Figure 5a plots the solution pH during Ni^2+^ removal. The initial solution pH was ~6.3. As the T400 came into contact with the aqueous solution, the pH rose rapidly to 7.1. The solution pH then declined slowly to 6.7 in 60 min before staying between 6.6 and 6.8 over 5 h. It reveals the capability of the calcined LDH to rapidly raise the solution pH at the initial stage of recovery. As stated in Equation (4), calcined LDH rapidly forms OH^−^ in aqueous solution, which increases the solution pH rapidly. This reveals the ability of calcined LDH to rapidly increase the solution pH in the initial stage of recovery. During the later stage of recovery, Ni^2+^ may be condensed with hydroxide ions, as the reaction of Equation (5) occurred, causing the solution pH to drop. Figure 5b–d plot the fitted curves of the adsorption kinetics of Ni^2+^. As shown in Figure 5b, the pseudo-second-order model (R^2^ = 0.9979) fits the experimental data better than the pseudo-first-order model (R^2^ = 0.9779), suggesting that the removal of Ni^2+^ is a chemisorption process [61,62]. To identify the rate-controlling steps of the removal process, intraparticle diffusion (Figure 5c) and liquid film diffusion (Figure 5d) models are used. The diffusion of adsorbates can generally characterize the adsorption of adsorbates through a solid–liquid interface through the external boundary layer formed on a solid surface (liquid film diffusion) or the diffusion of adsorbates within the adsorbent to adsorbed sites (intraparticle diffusion) or a combination of the two. In Figure 5c, simulated by the intraparticle diffusion model, the adsorption process can be divided into two stages. In the first stage, the adsorption rate (k_id, 1_ = 20.03 mg/g/min^0.5^) is relatively high. This stage corresponds to the diffusion occurring on the outer surface [62,63], and this stage has a rapid diffusion rate. The second stage, with a low adsorption rate (k_id, 2_ = 2.195 mg/g/min^0.5^), corresponds to the diffusion of Ni^2+^ inside the calcined LDH [63,64]. This stage yields a low R^2^ of 0.6591, indicating that the intraparticle diffusion does not describe the diffusion process well. The liquid film diffusion model in Figure 5d fits better than the intraparticle diffusion model, with an R^2^ of 0.9668 and a small y-intercept of −0.86, showing that liquid film diffusion is the rate-controlling step.

Figure 6 presents the adsorption kinetics of Cu^2+^ removal by T400. Figure 6a plots the Cu^2+^ concentration against time. The concentration of Cu^2+^ decreased rapidly within the first 5 min from the initial 75 mg/L to ~25.9 mg/L. Subsequently, it decreased relatively slowly to ~3 mg/L at 60 min and 0.3 mg/L at 300 min. The inset in Figure 6a plots the solution pH during Cu^2+^ removal. The initial solution pH was ~3.9. As the T400 came into contact with the aqueous solution, the solution pH rose rapidly to 5.4 and remained between 5.3 and 5.5 for 120 min. Between 120 and 300 min, the solution pH increased linearly with time from 5.4 to 6.5. Figure 6b–d plot fitted curves for the kinetics of the adsorption of Cu^2+^. As shown in Figure 6b, the pseudo-second-order model fits the experimental data better than the pseudo-first-order model. The liquid film diffusion model in Figure 6d fits better than the intraparticle diffusion model in Figure 6c, with an R^2^ of 0.9649 and a small y-intercept of −1.46, indicating that liquid film diffusion is the rate-controlling step.

The concentrations of Li^+^ and Al^3+^ during the recovery of Ni^2+^/Cu^2+^ were measured to determine the leaching of T400 as a function of time. Figure 7 plots the leaching concentrations of Li^+^ and Al^3+^ during the Ni^2+^ (Figure 7a) and Cu^2+^ (Figure 7b) recovery. Figure 7a shows that the Li^+^ concentration increased rapidly to ~0.9 mmole/L within 5 min of Ni^2+^ recovery, showing that Li^+^ was released from the calcined LDH rapidly. The concentration of Al^3+^ during the recovery of Ni^2+^ was 0.05−0.1 mmole/L. The rate of Li^+^ leaching in 5 min was faster than the rate of Ni^2+^ recovery, confirming Li^+^ leaching (Equation (4)) followed by Ni^2+^ recovery (Equation (5)). In the later stage, the Li^+^ concentration was stable. Figure 7b shows that Li^+^ concentration increased rapidly to ~0.8 mmole/L within 5 min of Cu^2+^ recovery. Al^3+^ was barely detected (<0.1 mg/L).

### 3.5. Morphologies of Ni/Cu-Containing Products and Derived Nano-Catalysts

Figure 8 presents microstructural observations of T400 after Ni^2+^ recovery and its derived nano-catalyst. Figure 8a displays the SEM surface image of T400 after Ni^2+^ recovery. The Ni-containing product observed in Figure 8a comprises dense filaments with a width of ~10 nm, uniformly covering the platelet surface. The elemental composition of T400 after the recovery of Ni^2+^/Cu^2+^ and H_2_ reduction as determined by SEM-EDS is presented in Table 2. As shown in Table 2, for the T400 after Ni^2+^ recovery for 5 h, EDS elemental analysis revealed a nickel content of 4.45 at.%. In addition, a trace amount, 0.37 at.%, of sulfur was detected on the Ni-containing product. The sulfur content decreased to very little intensity to detect after H_2_ reduction treatment for 3 h. Figure 8b–d display SEM and TEM observations of the Ni-containing product on T400 after H_2_ reduction treatment for 3 h. As shown in the SEM surface image of Figure 8b, the reduced Ni-containing product comprises nanoparticles with diameters of ~10 nm, densely and uniformly dispersed on the platelet surfaces. Figure 8c shows the TEM-HAADF image of the reduced Ni-containing product. The TEM-HAADF images revealed the Z contrast of the samples. Figure 8c shows bright nanoparticles densely dispersed on the platelet surfaces, revealing that the nanoparticles contain nickel. Figure 8d displays the HRTEM image of some nanoparticles. The nanoparticles have a d-spacing of 0.203 nm, corresponding to the Ni (111) plane.

Figure 9 shows observations of the microstructure of T400 after Cu^2+^ recovery and its derived nano-catalyst. The Cu-containing product (shown in Figure 9a) exhibits various morphologies, which can be classified into two types. One is nanoflower-like and is found at the edges of the platelets (left-hand image of Figure 9a); the other consists of nanoplatelet-like structures with a width of ~100 to 300 nm, formed on the platelet surface (right-hand image of Figure 9a). EDS elemental analysis (Table 2) reveals that the copper content was 5.36 at.%. The sulfur content of the Cu-containing product was 2.23 at.%, which was higher than that of the Ni-containing product. It decreased to 1.14 at.% after H_2_ reduction treatment for 1 h. The reduced Cu-containing product (shown in the SEM images in Figure 9b) after a reduction time of 1 h comprised nanoparticles of various sizes. As shown in the left-hand image of Figure 9b, relatively large particles (~35 nm in diameter) were primarily found at the edges of the platelets, with a few on the platelet surfaces. In the right-hand image of Figure 9b, relatively small nanoparticles with about 10–25 nm diameters are observed on the platelet surfaces. Figure 9c shows the TEM-HAADF image of the reduced Cu-containing product. Figure 9c shows bright nanoparticles dispersed on the platelet surfaces, revealing that the nanoparticles contain copper. The HRTEM image (Figure 9d) presents a nanoparticle with a diameter of ~15 nm. The nanoparticle has a d-spacing of 0.209 nm, corresponding to the Cu (111) plane.

### 3.6. Adsorption Isotherms

To investigate the interaction between the adsorbate and the adsorbent surface, adsorption equilibrium data were analyzed using Langmuir (Equation (10)) and Freundlich isotherm models (Equation (11)) as follows [41,60,65]:
Langmuir model:    1/Q_e_ = 1/(k_L_Q_m_C_e_) + 1/Q_m_(10)
Freundlich model:    lnQ_e_ = lnk_F_ + 1/n·lnC_e_(11)
where C_e_ (mg/L) is the concentration of adsorbate at equilibrium; Q_e_ (mg/g) is the amount of adsorbate adsorbed per unit weight of adsorbent at equilibrium; Q_m_ (mg/g) is the maximum adsorption capacity; k_L_ (L/mg) is the Langmuir constant related to adsorption energy; and k_F_ (mg/g·(L/mg)^1/n^) and n are Freundlich constants related to adsorption capacity and adsorption intensity, respectively. The Langmuir isotherm describes that all specific adsorption sites have homogeneous adsorption capacity to adsorbates, whereas the Freundlich isotherm model assumes that adsorption sites are heterogeneous [60,62,65]. Figure 10 presents the isotherms of adsorption of Ni^2+^/Cu^2+^ using T400. As shown in Figure 10a, the amount of Ni^2+^/Cu^2+^ adsorbed increased with initial concentration. At the maximum initial cation concentration of 600 mg/L, the removal capacities of Ni^2+^ and Cu^2+^ were 376.1 and 460.2 mg/g, respectively. Table 3 presents the isotherm adsorption parameters. As shown in Table 3, the Freundlich model fits better than the Langmuir model for both Ni^2+^ and Cu^2+^, suggesting that the active adsorption sites for Ni^2+^/Cu^2+^ of the calcined Li–Al LDH were heterogeneously distributed, and the recovery of metal cations might have occurred on multiple molecular layers [65,66]. The Freundlich constants (n) exceed 1 (Table 3), indicating that the isotherms are favorable [62,67].

### 3.7. Characterization of Ni/Cu-Containing Products

Figure 11 presents the characterization of the used T400 after the recovery of Ni^2+^/Cu^2+^ at various initial solution pH levels for 1 h. T400 was used to investigate the effect of the solution pH on Ni/Cu-containing products. The products are denoted as Ni-pH4.0, Ni-pH6.3, Ni-pH7.5 and Cu-pH3.9, Cu-pH5.5, and Cu-pH7.5, indicating the metal cations in the aqueous solution and the solution pH. Figure 11a presents GAXRD patterns of the Ni-pH6.3 and Cu-pH5.5 products. The Ni-pH6.3 product (the bottom pattern in Figure 11a) yields characteristic peaks that correspond to α–Ni(OH)_2_ (JCPDS 38–0715) at 2θ values of 11.35° and 22.75°. α–Ni(OH)_2_, with the chemical formula of Ni(OH)_2_·xH_2_O, is an LDH-like-structured compound that is always hydrated because water molecules are intrinsic to its structure [68,69]. The Cu-pH5.5 product (upper pattern in Figure 11a) yields a sharp diffraction peak at a 2θ of 12.75°, which is assigned to the (001) plane of posnjakite (Cu_4_(SO_4_)(OH)_6_·H_2_O, JCPDS 83–1410). Posnjakite is a hydrated copper sulfate mineral with a layered structure [70]. The GAXRD pattern of the Cu-pH5.5 precursor includes another broad peak with 2θ from ~8° to ~12°. This broad peak corresponds to Cu–Al–SO_4_ LDH (Cu_6_Al_2_(SO_4_)(OH)_16_·4H_2_O, JCPDS 29–0529). As the dotted line in Figure 11a shows, the regeneration of the characteristic peak of the Li–Al–CO_3_ LDH (002) plane is insignificant in one hour.

Figure 11b shows FTIR spectra of the products following the recovery of Ni^2+^/Cu^2+^ by T400 at various solution pH values. Figure 11b also compares the spectrum of T400-water (bottom curve, showing the spectrum of the T400 dipped in pure water for one hour before the FTIR test). The spectra of the Ni-pH4.0, Ni-pH6.3, and Ni-pH7.5 products include insignificant ν_O–H_ bands at around 3400 nm^−1^. The ν_O–H_ bands in the spectra of the Cu-pH3.9, Cu-pH5.5, and Cu-pH7.5 were significantly more extensive than that obtained in T400-water, perhaps because of the newly formed products (posnjakite and Cu–Al–SO_4_ LDH). The spectra of the Ni- and Cu-containing products also reveal the regeneration of the δ_H2O_ band at 1630 nm^−1^, consistent with the XRD results (Figure 11a), which reveal that the newly formed Ni/Cu-containing products contained water molecules. In the spectra of Cu-pH3.9 and Cu-pH5.5, a sharp band at 1115 cm^−1^ is attributable to the vibration of S–O (ν_S–O_) [71,72,73,74,75], indicating that the sulfate groups were intercalated into the newly formed phases. The ν_S–O_ band was absent in the spectrum of Cu-pH7.5, while the ν_3_, CO_3_^2−^ the band was present at 1380 cm^−1^. These results suggest that in the solution at a relatively high pH, carbonate groups, rather than sulfate groups, entered the interlayers because of the relatively high solubility of CO_3_^2−^ at a high pH [76], as well as the affinity of carbonate anions for intercalation into the interlayers of LDHs exceeding that of the sulfate anions [72,77]. In the spectra of the Ni-containing products, the ν_S–O_ band is relatively weak, indicating that the ability of the Ni-containing products to adsorb sulfate groups was insignificant. The weak adsorption of sulfate groups can be attributed to the substitution of some surface Al^3+^ for Ni^2+^, allowing some anions, such as SO_4_^2−^, CO_3_^2−^, and others, to enter the space between the Ni(OH)_2_ layers to maintain charge balance [78].

Figure 11c presents the Ni/Cu and S elemental contents of the Ni/Cu-containing products as determined by EDS. As shown in the left-hand panel of Figure 11c, the nickel content of the Ni-containing products increased significantly from 2.76 at.% to 5.17 at.% as the initial pH increased from 4.0 to 6.1, and then increased to 5.68 at.% at an initial pH of 7.5. These results suggest that the capacity to adsorb Ni^2+^ generally increased with initial solution pH. The copper content of the Cu-containing products (shown in the right-hand panel of Figure 11c) increased from 4.31 at.% to 5.16 at.% as the initial pH increased from 3.9 to 5.5, and then decreased as the initial pH of 7.5 (3.87 at.%). The copper content of the products decreased as the initial pH increased to 7.5 because some Cu^2+^ precipitated with hydroxide ions in solution at the high pH instead of forming Cu-containing products on the calcined LDH platelet surface. The sulfur contents of the Ni- and Cu-containing products on the calcined LDH platelet surface decreased as pH increased from 1.91 at.% at an initial pH of 4.0 to 0.94 at.% at an initial pH of 7.5 on the Ni-containing products. The sulfur contents in the Cu-containing products were relatively high at an initial pH of 3.9 and 5.5, while the sulfur content of the Cu-containing products at a pH of 7.5 was shallow, at only ~0.06 at.%. These results are consistent with the FTIR results in Figure 11b, which reveal that the Cu-containing products had a relatively high capacity to intercalate sulfate groups (ν_S–O_) at low pH. At a high pH (the Cu-pH7.5), carbonate groups were more likely to be intercalated into interlayers than sulfate groups.

Figure 11d plots the zeta potentials of Li–Al LDH, gibbsite, and their calcination products in weakly acidic water (pH 5.0) as a function of time. The changes of T400 in Ni^2+^/Cu^2+^-containing aqueous solution with time are also shown. As those compounds were immersed in the weakly acidic water, the zeta potential of LDH at 1 min was +10.6 mV; that of gibbsite was +16.6 mV; notably, that of the T400 was −16.7 mV; and that of the C-gibbsite was −1.7 mV. By comparing the zeta potential of the LDH with that of the T400, the results in Figure 11d show that calcination changed the surface from positively charged to negatively charged. Similar results were found for the gibbsite and C-gibbsite. Furthermore, the influence of calcination on zeta potential from positive to negative was much stronger on Li–Al LDH than on the gibbsite. The Li–Al LDH is composed of gibbsite sheets with Li^+^ in some of the vacant octahedral sites to form positively charged layers [16,17,18]. It indicates that the addition of Li^+^ makes the surface of the calcined product of Li–Al LDH more negative than that of the gibbsite. The zeta potentials of T400-Ni and T400-Cu at 1 min were +6.7 and +7.3 mV, respectively, indicating that T400 was positively charged when in contact with solutions containing Ni^2+^/Cu^2+^, perhaps due to the rapid surface bonding of Ni^2+^/Cu^2+^. The zeta potential of Li–Al LDH increased, and that of the gibbsite decreased over time, but both remained positively charged at 60 min. The zeta potentials of calcined LDH T400 in weakly acidic water and Ni^2+^/Cu^2+^-containing aqueous solutions varied little in 60 min (<2 mV).

### 3.8. Recovery Mechanism

Figure 12 schematic represents the recovery mechanisms of Ni^2+^ (Figure 12a) and Cu^2+^ (Figure 12b) on calcined Li–Al LDH. Figure 12a schematically depicts the mechanism of Ni^2+^ recovery on calcined Li–Al LDH. As shown in the left-hand panel of Figure 12a, during the initial stage of recovery, some of the surface oxygen species immediately reacted with water and released OH^−^, increasing the solution pH (Equation (4)). Some other surface oxygen species bonded to Ni^2+^, rapidly reducing the Ni^2+^ concentration (Equation (5)). The initially adsorbed Ni^2+^ attracted OH^−^ in solution electrostatically. In the later stage of recovery (right-hand panel of Figure 12a), the remaining Ni^2+^ gradually condensed with the high concentration of OH^−^ around the platelet surfaces, growing into condensates of α–Ni(OH)_2_. Figure 12b schematically depicts the mechanism of Cu^2+^ recovery on calcined Li–Al LDH. Similar to Ni^2+^, Cu^2+^ is primarily bound to surface oxygen species in the initial stage of recovery (as shown in the left-hand panel of Figure 12b). Cu^2+^ may be corrosive to calcined Li–Al LDH, especially at the platelet edges, so some Al^3+^ and OH^−^ were dissolved and released around the platelet surface. In the later stage of recovery (as shown in the right-hand panel of Figure 12b), Cu^2+^ precipitated or coprecipitated with Al^3+^, and the intercalation of sulfate groups accompanied this process into the metal hydroxide layers during the condensation.

## 4. Conclusions

This study established a method for recovering heavy metal cations from contaminated wastewater and converting the heavy metals into nano-catalysts. Li–Al–CO_3_ LDH was grown in situ on industrial aluminum lathe waste surface by immersing the waste in an alkaline aqueous solution that contained Li^+^ and Al^3+^. High-temperature calcination changed the surface properties and enhanced the ability to remove heavy metal cations from contaminated wastewater. The adsorption kinetic analysis revealed that the recovery of metal cations was a chemical reaction whose rate was controlled by external liquid film diffusion. This finding suggests that the surface bonding of the oxygen species with metal cations was an essential step in recovery, rapidly reducing the concentration of heavy metal cations. The Freundlich model fitted the adsorption isotherms, indicating that the adsorption was heterogeneous, perhaps because the condensed metal hydroxides provided new adsorption sites for heavy metal cations, allowing the condensed metal hydroxides to continue growing. The metal hydroxide products can be transferred into highly dispersed catalytic nanoparticles supported on Li–Al mixed oxide platelets by high-temperature hydrogen reduction treatment. The reduced nano-catalysts would be useful for specific applications such as the Cu–Ni two-stage catalytic reaction of ethanol steam reforming for hydrogen production. Other further applications include hydrocarbon reforming, dehydrogenation, hydrogenation (for the Ni catalyst), and hydrogenation of carbon–oxygen bonds into chemicals and fuels (for the Cu catalyst).

## Figures and Tables

**Figure 1 nanomaterials-13-01477-f001:**
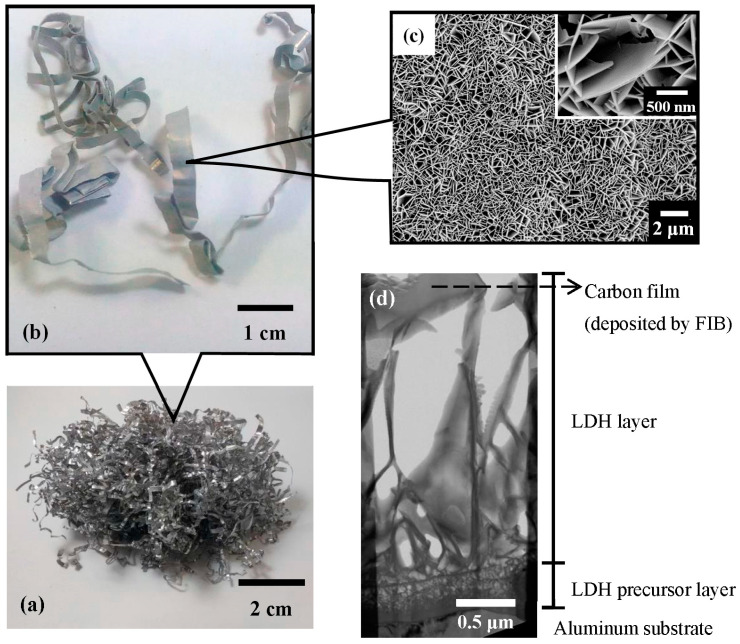
The macroscopic and microscopic views of the LDH/Al adsorbent material: (**a**) a photograph of industrial aluminum lathe waste (weight of 5 g), which would be employed as the substrates for the formation of Li–Al–CO_3_ LDH; (**b**) a photograph of some aluminum lathe waste strips after being coated with Li–Al–CO_3_ LDH, in which (**c**) SEM surface morphologies of the LDH are shown; and (**d**) TEM cross-sectional view of the LDH.

**Figure 2 nanomaterials-13-01477-f002:**
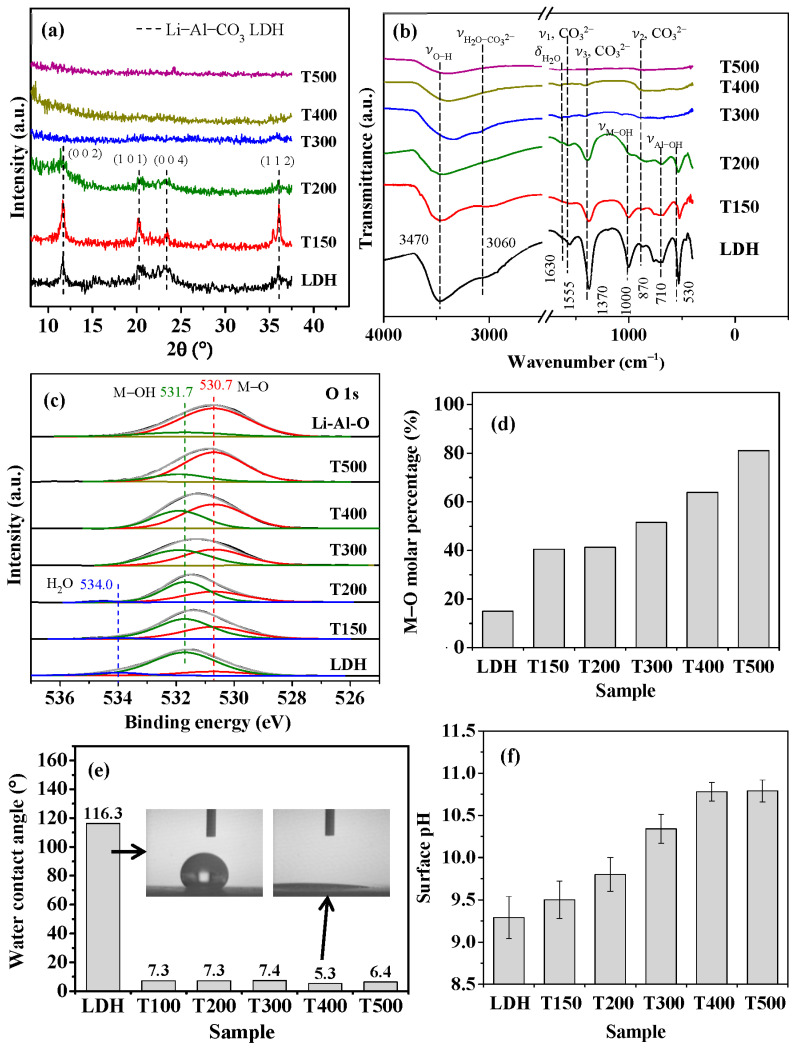
Surface characterization of LDH and calcined LDH with various calcination temperatures: (**a**) XRD patterns; (**b**) FTIR spectra; (**c**) O 1s XPS spectra; (**d**) M–O/M–OH molar percentage; (**e**) water contact angle; and (**f**) surface pH.

**Figure 3 nanomaterials-13-01477-f003:**
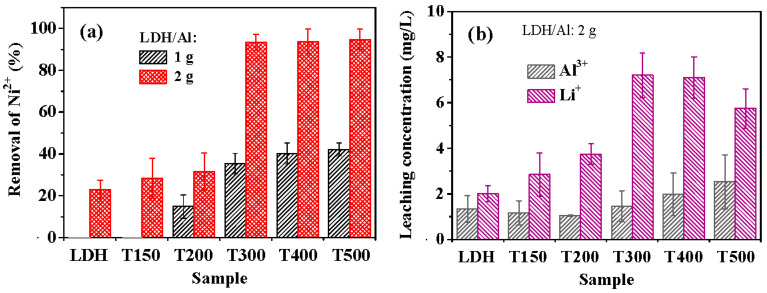
(**a**) Ni^2+^ removal efficiencies of LDH, T150, T200, T300, T400, and T500 from initial 75 mg/L Ni^2+^ aqueous solutions, indicating that using 2 g LDH/Al exhibits much better removal efficiencies than 1 g LDH/Al; (**b**) leaching concentrations of Li^+^ and Al^3+^ from the LDH, T150, T200, T300, T400, and T500 after Ni^2+^ removal experiments in aqueous solutions with initial 75 mg/L Ni^2+^. The time was 60 min for the Ni^2+^ removal experiments.

**Figure 4 nanomaterials-13-01477-f004:**
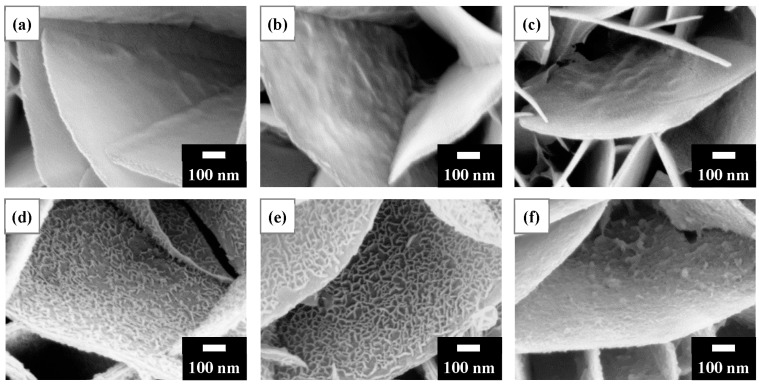
SEM surface morphologies after the recovery of Ni^2+^ for 1 min using (**a**) LDH and calcined LDH of: (**b**) T150; (**c**) T200; (**d**) T300; EUR T400; and (**f**) T500. As shown in (**d**,**e**), nanofilaments were observed on the platelet surface, especially T400 (as revealed in (**e**)). The initial Ni^2+^ concentration for the experiments was 75 mg/L.

**Figure 5 nanomaterials-13-01477-f005:**
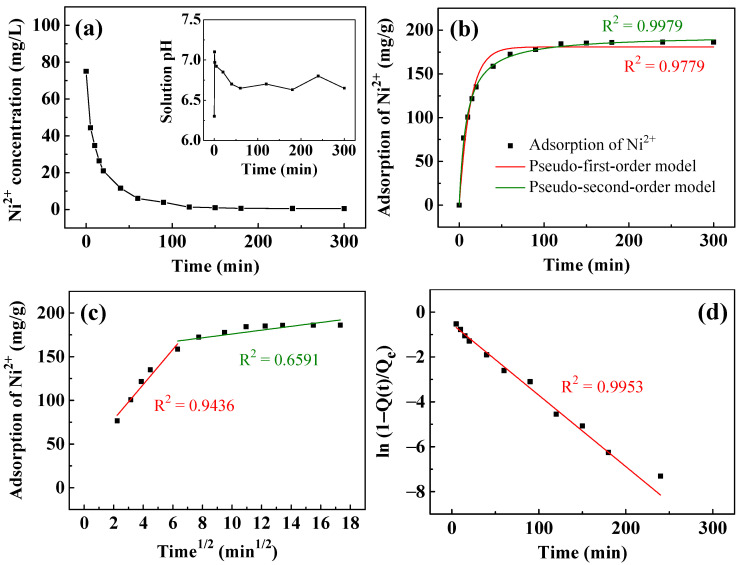
(**a**) The removal of Ni^2+^ and pH change (inset) as a function of time using T400. Kinetic fitting curves simulated by: (**b**) pseudo-first-order and pseudo-second-order; (**c**) intraparticle diffusion; and (**d**) liquid film diffusion models. The adsorbent dosage of T400 for this experiment was 0.4 g/L.

**Figure 6 nanomaterials-13-01477-f006:**
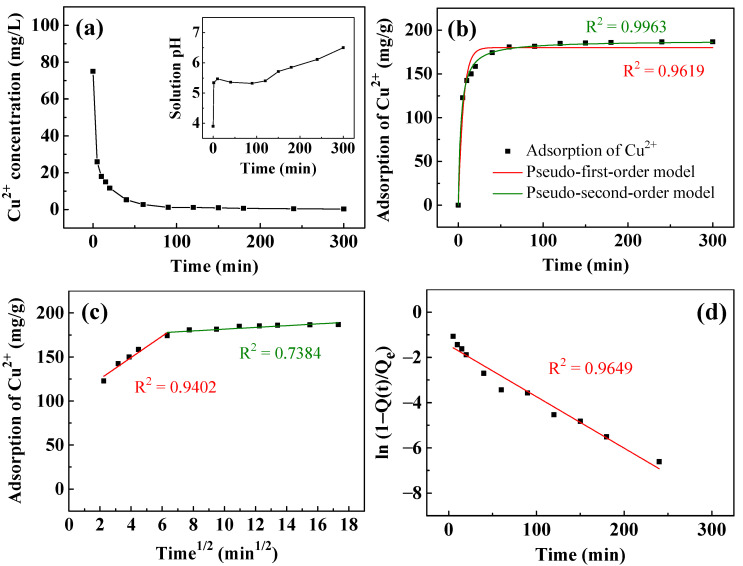
(**a**) The removal of Cu^2+^ and pH change (inset) as a function of time using T400. Kinetic fitting curves simulated by: (**b**) pseudo-first-order and pseudo-second-order; (**c**) intraparticle diffusion; and (**d**) liquid film diffusion models. The adsorbent dosage of T400 for this experiment was 0.4 g/L.

**Figure 7 nanomaterials-13-01477-f007:**
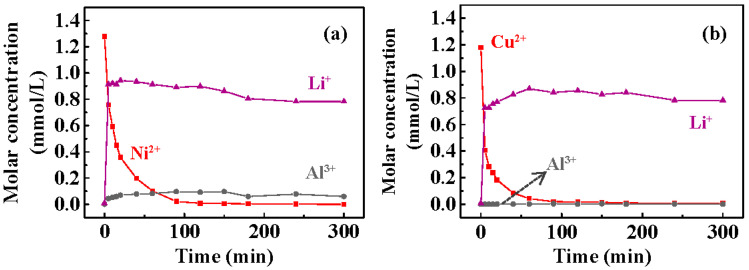
The leaching concentrations of Li^+^ and Al^3+^ from T400 as the removal of: (**a**) Ni^2+^ and(**b**) Cu^2+^. The Ni^2+^ and Cu^2+^ removal concentration data were re-plotted from Figure 5a and Figure 6a.

**Figure 8 nanomaterials-13-01477-f008:**
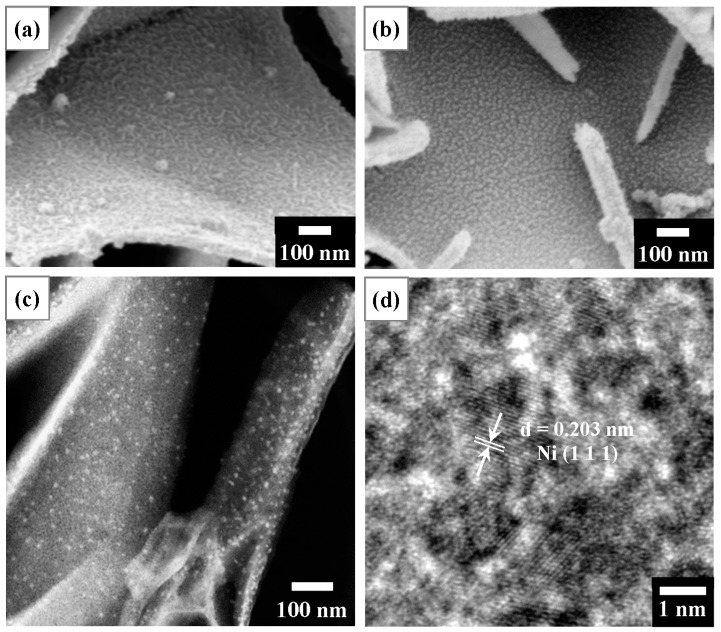
(**a**) SEM surface image of T400 after Ni^2+^ recovery for 5 h, indicating nanofilaments on the platelet surface; (**b**) SEM surface image of the sample of (**a**) after H_2_ reduction treatment, showing nanoparticles on the platelet surface; (**c**) TEM-HAADF observation of the sample of (**b**), showing bright nanoparticles; and (**d**) HRTEM image from the sample of (**c**), showing Ni (111) planes with d = 0.203 nm.

**Figure 9 nanomaterials-13-01477-f009:**
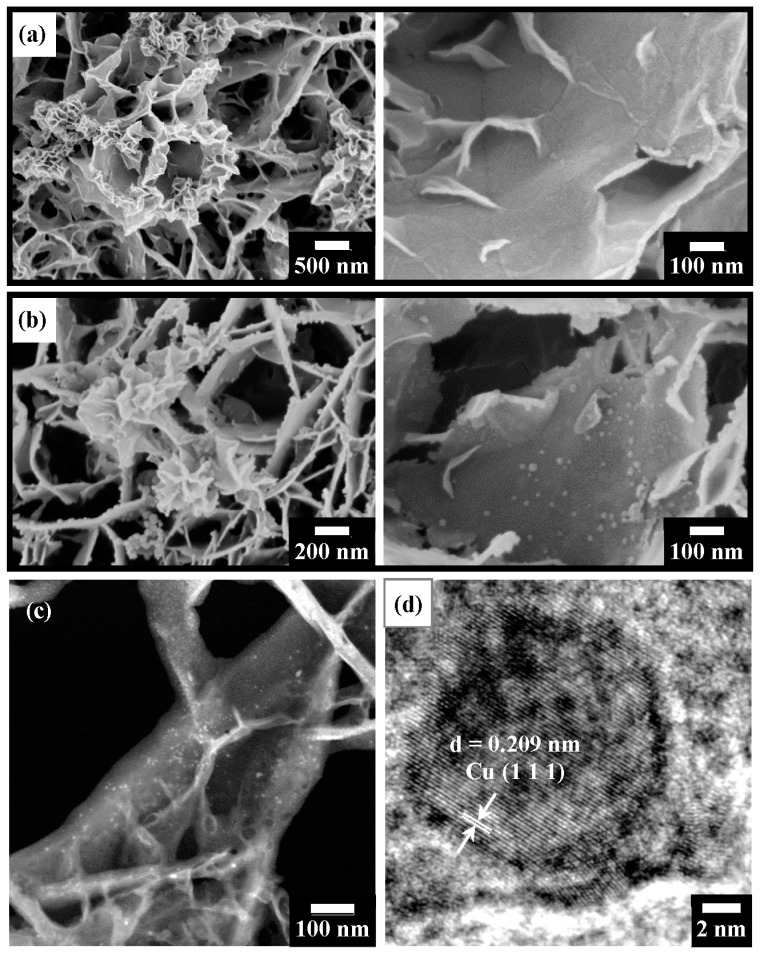
(**a**) SEM surface images of T400 after Cu^2+^ recovery for 5 h, showing nanoflower-like structures on platelet edges (left image) and nanoplatelets on platelet surfaces (right image); (**b**) SEM surface images of the sample of (**a**) after H_2_ reduction treatment, showing nanoparticles on platelet edges (left image) and on platelet surfaces (right image); (**c**) TEM-HAADF view of the sample of (**b**), indicating bright nanoparticles; and (**d**) HRTEM image from the sample of (**c**), showing Cu (111) planes in a Cu nanoparticle with d = 0.209 nm.

**Figure 10 nanomaterials-13-01477-f010:**
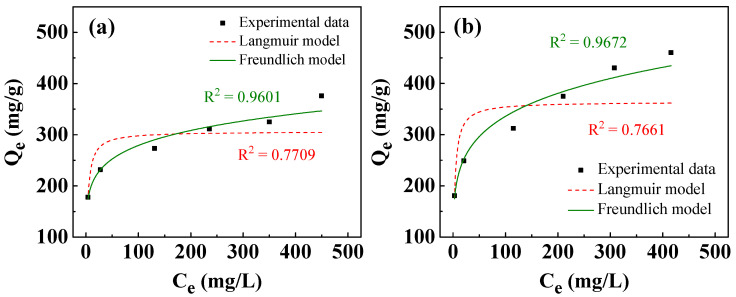
Adsorption isotherms fitted by Langmuir and Freundlich models for removal of: (**a**) Ni^2+^ and (**b**) Cu^2+^ using T400. The adsorbent dosage of T400 for the experiments was 0.4 g/L, with a one-hour immersion of the T400 in Ni^2+^/Cu^2+^-containing solutions.

**Figure 11 nanomaterials-13-01477-f011:**
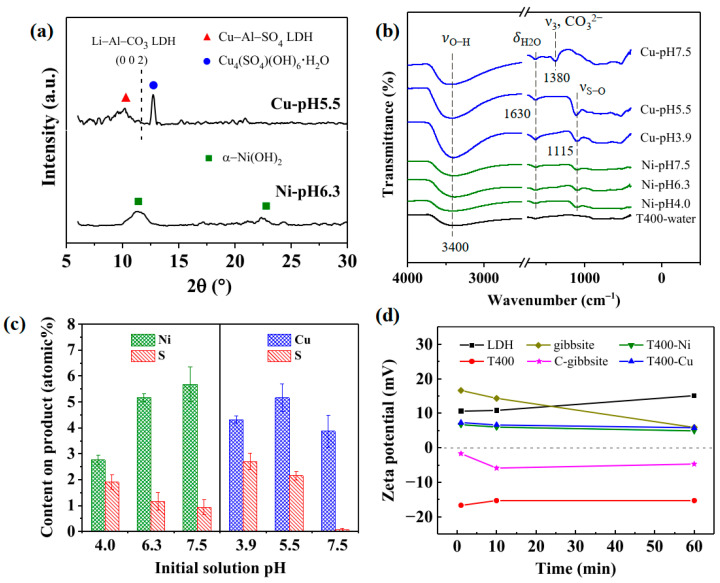
Characterization of the used T400 after the recovery of Ni^2+^/Cu^2+^ at various initial solution pH levels for 1 h: (**a**) XRD pattern; (**b**) FTIR spectra; (**c**) EDS elemental content on products; and (**d**) Zeta potentials of Li–Al LDH, gibbsite, and their calcination products in weakly acidic water (pH 5.0) as a function of time. The changes of T400 in Ni^2+^/Cu^2+^-containing aqueous solution with time are also shown. The initial Ni^2+^/Cu^2+^ concentration for the experiments was 75 mg/L.

**Figure 12 nanomaterials-13-01477-f012:**
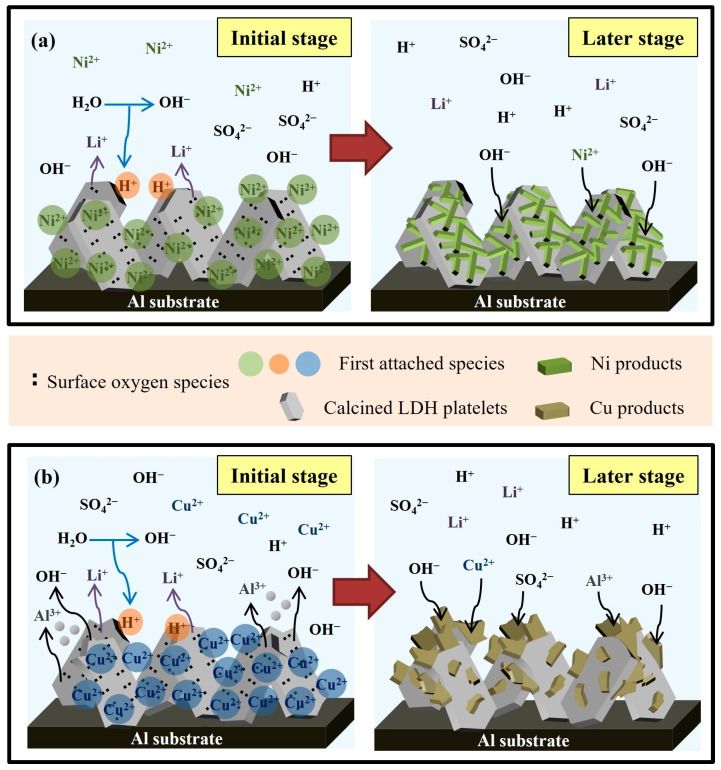
Schematic representations of the recovery mechanisms of (**a**) Ni^2+^ and (**b**) Cu^2+^ on calcined Li–Al LDH.

**Table 1 nanomaterials-13-01477-t001:** Kinetic parameters and adjusted coefficients of determination of the adsorption kinetic models for Ni^2+^/Cu^2+^ removal using T400.

		Ni^2+^	Cu^2+^
Pseudo-first-order model	k_1_ (mg/g/min)	0.0779	0.1729
	Q_e_ (mg/g)	180.8	180.1
	R^2^	0.9779	0.9619
Pseudo-second-order model	k_2_ (mg/g/min)	0.000596	0.001730
	Q_e_ (mg/g)	194.4	188.0
	R^2^	0.9979	0.9963
Intraparticle diffusion model	K_id, 1_ (mg/g/min^0.5^)	20.03	12.14
	C (mg/g)	38.13	100.9
	R^2^	0.9436	0.9402
	k_id, 2_ (mg/g/min^0.5^)	2.195	0.997
	C (mg/g)	154.1	171.6
	R^2^	0.6591	0.7384
Liquid film diffusion model	k_fd_ (min^−1^)	0.0318	0.0228
	C (mg/g)	−0.530	−1.460
	R^2^	0.9953	0.9649

**Table 2 nanomaterials-13-01477-t002:** Elemental composition of T400 after the recovery of Ni^2+^/Cu^2+^ and H_2_ reduction as determined by SEM-EDS.

	Elemental Composition (Atomic %) as Determined by SEM-EDS			
	O	Al	Ni	S
After Ni^2+^ recovery for 5 h	73.00	22.18	4.45	0.37
After H_2_ reduction for 3 h	61.70	33.33	4.97	--
	O	Al	Cu	S
After Cu^2+^ recovery for 5 h	72.37	20.05	5.36	2.23
After H_2_ reduction for 1 h	65.21	27.74	5.92	1.14

**Table 3 nanomaterials-13-01477-t003:** Adjusted coefficients of determination of Langmuir and Freundlich isotherm models for Ni^2+^/Cu^2+^ removal using T400.

		Ni^2+^	Cu^2+^
Langmuir model	Q_m_ (mg/g)	306.7	363.6
	k_L_ (L/mg)	0.3432	0.3508
	R^2^	0.7709	0.7661
Freundlich model	k_F_ (mg/g·(L/mg)^1/n^)	143.8	146.0
	n	6.942	5.528
	R^2^	0.9601	0.9672

## Data Availability

The data that support the findings of this study are available on request from the corresponding author.

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
