# Peer review of "The Effect of Lithium Ion Leaching from Calcined Li–Al Hydrotalcite on the Rapid Removal of Ni^2+^/Cu^2+^ from Contaminated Aqueous Solutions"

_nanomaterials, 2023, doi:10.3390/nano13091477_

Round 1

Reviewer 1 Report

The article is devoted to the study of the possibility of using Layered double hydroxide (LDH) calcined framework adsorbent for rapid removal of heavy metal cations from plating wastewater. Li−Al−CO3 LDH was synthesized on an aluminum lathe waste frame surface to prepare the sorbent. The effect of changing surface functional groups and leaching of lithium ions on surface properties and adsorption capacity is also considered.

The article is very good, scientific, relevant, but I cannot agree with one conclusion of the authors, the processing of experimental data using the intraparticle diffusion model of adsorption is not presented correctly (Fig. 5 (c) and Fig 6 (c) and data in Table 1). It is quite clear that the experimental data should be divided into two linear sections, each responsible for its own adsorption mechanism and adsorption rates with other parameters related to different linear regions will also be different. Therefore, the data presented for this model in Table 1 must be changed! A detailed description and analysis of such a situation is given in many works, for example, [Chun-Rong Lin, et all, Nanomaterials 2022, 12, 376. https://doi.org/ 10.3390/nano12030376 ,  Hengli Xiang et all,  Nanomaterials 2021, 11, 330. https://doi.org/10.3390/nano11020330]

 in addition, equations 1, 2, 9 must be written in a form that excludes their double understanding (for example, using brackets)

In general, I think that the article is worthy of being published in the journal Nanomaterials after correction

Reviewer 2 Report

The manuscript deals with the rapid removal of Ni2+/Cu2+ From Contaminated Aqueous Solutions using calcined Li-Al Hydrotalcite.

Line 41-43: It should mention. https://onlinelibrary.wiley.com/doi/full/10.1002/jctb.6912

Line 59-60: Why do you think the use of calcination is appropriate when it raises economic costs especially nowadays.

Indicate the novelty of this work at the end of the introduction.

It can be helpful later https://www.mdpi.com/2313-4321/2/4/20.

Line 115: Indicate purity.

Line 111: Explain the choice of these calcination temperatures.

Line 138: Samples should be coated with gold and palladium prior to actual measurement. 

Line 150: Indicate the number of scans for FTIR measurements.

Line 231: Figure 2b: At higher temperatures, dehydroxylation occurs and the transition to the unactive form of hydrotalcite. Wouldn't it be wise to activate the hydrotalcite further with some alkaline activator to create a geopolymer for example, or other mineral phases for contaminant removal?

Line 309: Measurement deviations should be given. How many times have the measurements been repeated?

Line 385: SEM is mostly suitable to be used in micrometer and TEM nanometer compositions. Some of the images seem to be unclear. 

Line 565: Conclusion and also and future research perspectives could be given. 

Reviewer 3 Report

The manuscript «Effect of Leaching of Lithium Ions from Calcined Li−Al Hydrotalcite on Rapid Removal of Ni2+/Cu2+ From Contaminated Aqueous Solutions» is devoted to the study of the mechanism concerning the quick uptake of heavy metal cations from wastewater.

The material of the article has a high novelty, scientific and practical significance. I believe that the manuscript can be published in the journal Nanomaterials after taking into account the following comments, the solution of which, in my opinion, will significantly improve the quality of the article.

1. The literature review in the manuscript is presented in sufficient quality and detail, especially with regard to the objects of study. Nevertheless, I advise you to add a few suggestions regarding the general requirements for adsorbents, the level of toxicity of the studied metals in drinking water, as well as known methods for the regeneration and removal of heavy metal concentrates.

2. In this work, the products after adsorption of Ni2+/Cu2+ are characterized in sufficient detail. However, more attention should be paid to the characterization of samples after reduction in a hydrogen flow, since they are the most interesting for further use in catalysis. In particular, XRD results showing the presence of metal phases should be added; EDX results showing the change or absence of sulfur and oxygen in the samples after reduction, and, if possible, the results of TG/DSC analysis of the samples before drying.

3. In the conclusions section, I would like to see examples of specific processes/applications where these catalysts could be used.

4. In Eq. 2, the analysis of the dimensions of the quantities used in the right side of the equality (taking into account lines 122 and 124) does not give the unit Qt (mg/g).

5. l.126 Was the adsorbent weighed after drying, or was the weight of the sample before drying used to calculate adsorption? How significant are the changes in the mass of the sorbent before and after drying?

6. l.194 Show XRD patterns with a wider range of 2theta to prove the amorphousness of the samples after drying.

7. Fig. 5a, inset. Decrease the pH range (e.g. to 6.25-7.25) for more representativeness.

8. l.318 Describe also the reason for the subsequent decrease in pH (refs to Eqs. 4, 5)

9. l.354-359, Fig 7. Present concentrations in units of mmol/L or mM.

10. l.397. Give the results of the EDX analysis in the form of a table. How does the content and chemical nature of sulfur change during reduction?

11. l.444 Describe in the experimental part how the pH values were varied.

12. l.548 In the figure caption, correct "b, c" to "a, b".

Round 2

Reviewer 1 Report

The authors have done a lot of work to improve the quality of the article. Now the article can be published in the journal Nanomaterials.

Reviewer 2 Report

It was improved and can be accepted. 

Reviewer 3 Report

The authors responsibly approached the correction of the manuscript and, in our opinion, significantly improved the quality of the article. Therefore, I believe that the manuscript can be accepted for publication as is.

However, in 2 questions, the authors did not quite understand me correctly.

Point 4 implies that to get Qt (mg/g) you need to use V(L)

Point 7 Just had to change the pH range on the graph to increase it

However, these remarks are minor.